# Possibilities of Rock Processing with a High-Pressure Abrasive Waterjet with an Aspect Terms to Minimizing Energy Consumption

**DOI:** 10.3390/ma16020647

**Published:** 2023-01-09

**Authors:** Grzegorz Chomka, Maciej Kasperowicz, Jarosław Chodór, Jerzy Chudy, Leon Kukiełka

**Affiliations:** 1Branch of the Koszalin University of Technology in Szczecinek, Koszalin University of Technology, Waryńskiego 1 Street, 78-400 Szczecinek, Poland; 2Faculty of Mechanical Engineering, Koszalin University of Technology, Racławicka 15-17 Street, 75-620 Koszalin, Poland

**Keywords:** high-pressure waterjet, high-pressure abrasive waterjet, rock processing, abrasive waterjet (AWJ)

## Abstract

The paper concerns the application a high-pressure abrasive waterjet (AWJ) for cutting the most commonly used rock materials such as granite, limestone, basalt and marble. Based on the analysis of the literature, the influence of parameters on the specific energy E_v_, specific energy of cutting E_r_ and specific energy of intersection E_a_ was determined. Experimental studies were carried out on a laboratory test stand in accordance with a five-level rotating experiment plan. The results of the research were subjected to statistical processing, obtaining regression equations. The influence of the pressure and diameter of the abrasive waterjet on the energy consumption of rock cutting was tested. The effect of the stream power, feed speed and pressure on the cutting depth with the AWJ was also determined. The data obtained made it possible to evaluate the machinability of the rocks as a function of the power of the jet. These analyses were supplemented with charts illustrating the influence of the most important technological parameter of the cutting process, which is the feed speed. The presented results provide answers to the energy and time requirements for efficient cutting with the AWJ of frequently used rock materials.

## 1. Introduction

High-pressure water treatment technology has been developed for more than 50 years. At that time, articles were published on the use of high-pressure waterjets to cut through various materials [1,2,3,4,5,6] and high-pressure waterjets with additives designed to increase processing efficiency. Ice particles are added to the high-pressure waterjet [7,8,9,10], along with solidified carbon dioxide particles CO_2_ [11,12], or garnet, corundum or olivine [13,14,15,16]. High-pressure waterjets with dry ice admixtures CO_2_ are mostly used for cleaning surfaces [17]. On the other hand, high-pressure waterjets with harder admixtures are used to cut and separate mainly flat titanium materials [18,19,20], heat-resistant steels [21,22,23], austenitic steels [24,25,26,27,28], aluminum alloys [29,30,31,32,33], copper alloys [34,35], plastics reinforced with different types of fibers [36,37,38,39] and glass or ceramics [40,41,42]. There are also reports of research into the use of high-pressure waterjets for cutting through rock materials.

Aydin et al. [43] used granite waste as an abrasive added to a waterjet used to cut marble. It is revealed that the granite particles show similar performance with garnet in terms of the cutting width (granite: 2.10 mm and garnet: 2.21 mm) and the surface roughness (granite: 5.2 μm and garnet: 4.59 μm). It is determined that a lower cutting depth (67% of the cutting depth produced by garnet, 15.62 mm) is obtained with the granite particles. Additionally, it is concluded that garnet produces lower kerf angles (70% and 79% of the kerf angle-access and kerf angle-exit produced by the granite particles, respectively). Finally, it can be noted that the granite particles can be effectively used in the cutting of marble and other rocks with similar density and hardness.

Karakurt et al. [44] have identified the main relevant process factors affecting the depth of granite cutting. It was revealed that the sliding speed was the most important process parameters affecting the depth of granite cutting. In addition, it was found that the depth of the cut and surface quality were strongly influenced by the grain size and its boundaries with the surrounding grain. In addition, consistent relationships were observed between some of the physical and mechanical properties of granite (for example, water absorption, microhardness, specific bulk density and uniaxial compressive strength) and depth of the cut.

Barabas i Florescu [45] presented a method for reducing crack formation during hydro-jet cutting of marble using statistical analysis. A thorough study of the behavior of marble under the water-abrasive jet and the behavior of the entire process in processing brittle materials was conducted. Experimental results confirmed the hypothesis that statistical analysis is a procedure that can lead to a reduction in the number of cracks. The reduction in the number of cracks is achieved by using low pressure, a mini-minimum distance from the work surface and a small nozzle internal diameter.

In paper [46], Huang et al. presented the results of a study of cutting parameters on granite cutting performance. The mass flow rate of the abrasive was found to be proportional only to water pressure, and the effect of the other cutting parameters was insignificant. In addition, the increase in the water pressure is associated with the increased crack width and the reduced crack cone. The width of the cut decreases as the nozzle feed speed increases, resulting in a significant increase in the cone of the cut as the nozzle feed speed increases.

The scope of research on the application of high-pressure waterjets is very broad. Some authors [47] present the results of studies of important rock properties affecting the recycling of abrasives in granite cutting. In contrast, others [48,49,50,51,52] are conducting research on the disintegration of abrasive materials in waterjet processing. Although there has been tremendous recent progress in the development of new machinery, equipment and other technical devices that make waterjet processing more efficient, there is still a lack of comprehensive studies outlining its application to rock cutting. At the same time, no studies capturing simultaneously the results of research on the intensification of rock processing using a pure waterjet as well as a water-abrasive jet were also presented.

The purpose of the study is to determine the effect of process parameters such as pressure, water nozzle internal diameter and feed speed on cutting efficiency. The first two parameters determine the water output and the power of the jet, and the third determines the jet erosion time per unit volume of material. Their interdependence, using appropriate evaluation indicators, allows to determine the energy intensity of processing and directions for its minimization. This article therefore responds to existing research gaps.

## 2. Materials and Methods

### 2.1. Cutting Efficiency

There have been many attempts to mathematically represent the efficiency of abrasive cutting, which is commonly referred to as the cutting depth h. One of the first relationships is the M. Hashish equation [53]:(1)$$h=\frac{\sqrt{2\cdot {\dot{m}}_{a}\cdot v_{a}^{2}}}{8\cdot R_{e}\cdot v_{p}}+\frac{2\cdot {\dot{m}}_{a}\cdot \left(1-c\right)\cdot v_{a}^{2}}{\pi \cdot v_{p}\cdot \varepsilon \cdot d_{j}},\;\left[\mathrm{mm}\right],$$
where:

$\dot{m_{a}}$ = abrasive flow rate, [g·s^−1^],

*v_a_* = grain speed, [mm·s^−1^],

*R_e_* = yield stress, [MPa],

*v_p_* = feed speed, [mm·s^−1^],

*c* = a constant that determines the portion of the abrasive mass flow involved in cutting,

*ε* = specific energy of plastic deformations, [kJ],

*d_j_* = nozzle internal diameter, [mm].

It results from a two-way model of the cutting process based on the interpenetration of fatigue cutting at small and large grain impact angles. Unfortunately, this equation does not take into account the size of the abrasive particles or, more importantly, the feed speed. This makes it applicable only to plastic materials with 30–40% confidence in the representation of the real state [53].

The empirical relationship created by a group of Japanese scientists [54] more accurately depicts the actual cutting process because it already takes into account most of the parameters that determine its effectiveness. The initial factors of the equation are the power output, feed speed and distance of the nozzle from the workpiece while the type of workpiece material, type of abrasive and the ratio of abrasive and water output were classified as constant factors conditioning the specifics of the process. The above considerations apply to single-pass cutting, so this parameter is not included in the final form of the equation:(2)$$h=k_{1}\cdot \frac{N_{j}}{v_{p}\cdot l}+k_{2}\cdot \left(\frac{{\dot{m}}_{a}}{{\dot{m}}_{w}}\right)\cdot \frac{N_{j}}{v_{p}\cdot l},\;\left[\mathrm{mm}\right],$$
where:

*k*_1_, *k*_2_ = constants determined by the type of rock and abrasive,

*N_j_* = jet power, [kW],

*L* = distance of the nozzle from the surface of the object, [mm],

${\dot{m}}_{a}$ = abrasive flow rate, [g·s^−1^],

${\dot{m}}_{w}$ = water flow rate, [g·s^−1^].

The following equation developed by Rehbinder [55], in turn, makes the eroding parameters dependent on the pressure p and the internal diameter of the water nozzle *d_w_*, the feed speed *v_p_*, the numbers of passes *n* and the parameters of the treated granite, i.e., erosion resistance expressed as the ratio of average grain size to compressive strength *l/R_c_* and threshold pressure *p_th_*:(3)$$h=f\left\{d_{w}\left(\frac{p}{p_{th}},\frac{R_{c}\cdot p\cdot n}{\mu \cdot l\cdot v_{p}}\right)\right\},\;\left[\mathrm{mm}\right]$$

Another relatively simple relationship [56,57], applicable to single cuts, takes into account only three basic technological parameters of the cutting process:(4)$$h=k\cdot \frac{p^{a}\cdot d_{w}^{b}}{v_{p}^{c}},\;[\mathrm{mm}].$$

However, by appropriately selecting factors *a*, *b*, *c* with values derived from, among other things, the type of material to be processed, the abrasive used, or the abrasive output, they reflect the course of the water-abrasive cutting process with great confidence [56,57].

Based on the analysis of the above relationships, it can be concluded that the most significant factor affecting the efficiency of cutting with a high-pressure abrasive jet is its power *N_j_*, represented sometimes in entangled form as the product of the pressure and the internal diameter of the water nozzle. An important factor is also the feed speed, which determines the actual time the jet affects the surface of the material at a given point. Therefore, these quantities should be given priority in further considerations and experiments carried out.

There are many ways to comparatively represent the energy intensity and energy efficiency of cutting through rock materials with a high-pressure abrasive waterjet. This can be used most often with the specific energy of machining *E_v_*, which is an illustration of the energy expenditure required to remove a unit volume of machined material:(5)$$E_{v}=\frac{E_{j}}{V_{e}},\;\left[\mathrm{kJ}\cdot {\mathrm{mm}}^{-3}\right],$$
where:

*E_j_* = jet energy, [kJ],

*V_e_* = eroding volume, [mm^3^].

In addition to it, you can also meet the specific energy of the intersection *E_a_* expressing the energy required to make a cut with a unit area in the plane of motion of the working tool (water-abrasive jet), expressed by the ratio of power to the feed speed and the depth of the cut:(6)$$E_{a}=\frac{N_{j}}{h\cdot v_{p}}=\frac{E_{j}}{h\cdot l},\;\left[\mathrm{kJ}\cdot {\mathrm{mm}}^{-2}\right],$$
where:

*N_j_* = jet power, [kW],

*H* = eroding depth, [mm],

*v_p_* = feed speed, [mm·s^−1^],

*E_j_* = jet energy, [kJ],

*L =* length of cut, [mm].

The specific energy of cutting *E_r_* is also used to describe the amount of energy required to make a cut of unit depth:(7)$$E_{r}=\frac{E_{j}}{h},\;\left[\mathrm{kJ}\cdot {\mathrm{mm}}^{-1}\right],$$

The ace machining efficiency index is the ratio of the power of the jet to the unit depth of the cut obtained:(8)$$a_{s}=\frac{N_{j}}{h},\;\left[\mathrm{kW}\cdot {\mathrm{mm}}^{-1}\right],$$

At this point, it is important to note the close correlation between the last three indicators. Taking into account that the feed speed is the ratio of the length of the obtained cut to the duration of the cut, the specific energy of the cut can be represented as follows:(9)$$E_{a}=\frac{a_{s}}{v_{p}}=\frac{E_{r}}{l},\;\left[\mathrm{kJ}\cdot {\mathrm{mm}}^{-2}\right],$$

The use of four different comparative energy intensity assessment indicators can create ambiguities and distortions in the evaluation of results. Therefore, they were subjected to the following evaluation:–specific energy (*E_v_*_,_ kJ·mm^−3^) is dependent on the volume of the groove, which, with the decreasing depth of the cut, increases with multiple cuts or a large distance of the nozzle from the workpiece,–machining efficiency index does not take into account the temporal parameters of the process by which its comparison requires results obtained at a constant feed speed,–the specific energy of cutting *E_r_*, kJ·mm^−1^ may vary depending on the length of cuts or their duration,–the specific energy of intersection *E_a_*, kJ·mm^−2^ by capturing energy as a relationship between the power of the jet and the feed speed, eliminates the above disadvantages and fully captures the time-energy aspects of machining.

### 2.2. Materials

The objects of the research described in this article are mineral materials and primarily rocks. The division of rocks by geological origin is shown in Table 1. Of the contained minerals, representatives of all rock types were included in the study because of the possibility of comparing the effectiveness of the high-pressure abrasive jet treatment of materials that clearly differ in their different structures and properties. These include granite, limestone and marble. In addition, complementary studies to expand insights into rock processing included basalt and sandstone.

Granite has a full crystalline structure and a texture that is disorderly and massive. It is grayish-white, grayish-yellow or pinkish-red in color. The main components of granite are quartz, orthoclase (potassium feldspar) and dark components (hornblende and augite). It is found in the Sudeten Mountains. Granite is characterized by its high strength. The most durable are fine-grained granites with a large amount of evenly distributed quartz and a small amount of mica. Granite from the Strzegom quarry, illustrated in Figure 1, with the following properties, was used for the study:

Technical specifications of granite:–specific density         -*ϱ* = 2.65 Mg·m^−3^,–compressive strength       -*R_c_* =160–170 MPa,–abrasiveness in Deval’s drum    -*M_DE_* = 1.7–5.5%–Mohs hardness         -*MH* = 6–7.

Limestone is a rock with very diverse origins and properties. Very numerous are organic limestones made up of organic debris of various sizes and inorganic masses of calcium carbonate. When organic remains are recognizable to the naked eye, they are said to be biomorphic structures. Fine-grained limestones with massive textures are called compacted limestones. In addition, there are limestones of chemical origin: oolitic (made up of small, adherent balls of calcite), pelitic (a uniform, compact mass of calcium carbonate).

Technical data of limestone from Morawica quarry are shown in Figure 2:–specific density         -*ϱ* = 2.75 Mg·m^−3^,–compressive strength       -*R_c_* =92–125 MPa,–abrasiveness in Deval’s drum    -*M_DE_* = 4.3–23.5%–Mohs hardness         -*MH* = max. 3.

Marble is the name for rocks formed by the re-crystallization of sediments containing calcite or other carbonates, deep in the earth at high temperatures and under considerable pressure. It consists of equal grains of CaCO_3_, which are closely related and form a homogeneous mass. Marble has a medium-crystalline structure and is extremely resistant to weathering. It is found in the Sudetenland and is colored pink-green, gray, greenish or white. Marbles from the deposit in the area of Stronie Śląskie are called Green and White Marianna and have a spotted-white color while those mined near Slavniovice are white, gray or greenish.

The marble used in the study was Marianna white marble, a view of which is included in Figure 3, with the following parameters:–specific density         -*ϱ* = approx. 2.72 Mg·m^−3^,–compressive strength       -*R_c_* =52–58 MPa,–abrasiveness in Deval’s drum    -*M_DE_* = 2.7–7.5%–Mohs hardness         -*MH* = 3.

In addition, samples of sandstone and basalt, i.e., rocks with the highest and lowest strength parameters, were tested for comparative purposes.

Analysis of the cutting results of these rocks, which have very different properties due to their different geological origins, will show a complete picture of the effectiveness of high-pressure waterjet processing.

### 2.3. Methods and Test Rig 

The research was carried out using an abrasive waterjet cutting system from the US company Ingersoll Rand of the Hydroabrasives type. This device (Figure 4) is derived from the “Streamline” system. The most essential component of the system is the Ingersoll Rand Streamline high-pressure pump with a capacity of approximately 30 kW and a maximum pressure of 379 MPa and a water flow of approximately 4.5 dcm^3^·min^−1^. It is a double-acting reciprocating piston pump, hydraulically powered.

## 3. Results and Discussion 

In the analysis of hydraulic parameters, it is necessary to take into account such basic process factors as water pressure p and the internal diameter of the water nozzle *d_w_* and the resulting power of the jet *N_j_*. Knowing the influence of these parameters allows to evaluate the energetic aspects of the hydroabrasive cutting of rock materials.

Studies have shown that of the selected rocks, only two succumb to being cut by a stream of clean water. They are sandstone and granite. These rocks are very different from each other. What they have in common is that they are composed of many grains clumped together. The results are shown in Figure 5.

The conducted tests showed a linear dependence of the increase in cutting depth on the jet pressure regardless of the type of material being cut. The linear mapping is very accurate as the correlation coefficients are not lower than *R* = 0.985. The graphs in Figure 5 show that as the nozzle internal diameter increases, the erosion potential of the jet grows stronger. As the pressure increases, the directional coefficients of the straights change from a = 0.011 mm/MPa for the nozzle internal diameter *d_w_* = 0.13 mm to *a* = 0.45 mm/MPa for a *d_w_* = 0.34 mm nozzle internal diameter.

The proportional relationship between the pressure and the depth of the cut is also confirmed by the graphs in Figure 6, which show the results of cutting through rock with an abrasive waterjet. In this case, all materials are eroded, and the cutting efficiency is much higher than with a pure waterjet.

The highest cutting depths were obtained for sandstone. A 200 mm thick specimen was cut through, for the smallest nozzle with an internal diameter of *d_w_* = 0.13 mm at a maximum pressure of *p* = 350 MPa, and for larger ones at much lower pressures. For a nozzle with an internal diameter of *d_w_* = 0.34 mm, the sample was already cut at *p* = 200 MPa.

The most difficult to machine is hard and fine-grained basalt with a compressive strength of *R_c_* ≈ 230 MPa (Figure 7). Similarly resistant to such treatment are limestone and marble (*R_c_* ≈ 85–105 MPa). Slightly better workable is granite, characterized by higher strength parameters than the two previously mentioned rocks (*R_c_* ≈ 110–140 MPa) and, of course, sandstone with very low parameters *R_c_* ≈ 25–40 MPa. This is due to the multicomponent structure of these minerals, which determines the specific mechanism of water-abrasive erosion. This is a mechanism similar in its course to that occurring in the processing of concrete and involves the leaching of soft components of these rocks (mica) and the lifting of hard particles (quartz).

According to previous considerations and conclusions, the power should most fully reflect the effect of the hydraulic parameters on the effectiveness of the cutting rock materials because it fully captures aspects of the dynamics of the abrasive waterjet. The power of the jet can be described as the product of the mass flow rate of the liquid and the square of its velocity according to the following expression:(10)$$N_{j}=\frac{{\dot{m}}_{w}}{2}\cdot v_{w}^{2},\;\left[\mathrm{kW}\right].$$

It can also be expressed as the product of volumetric water output and its pressure according to the relationship:(11)$$N_{j}=V_{w}\cdot p,\;\left[\mathrm{kW}\right].$$

The nature of these relationships is best illustrated by the dependence of the depth of the cut of limestone, granite and marble on the power of the jet shown in Figure 8, Figure 9 and Figure 10. The results of the study are illustrated by nonlinear equations. The accuracy of this mapping is evidenced by correlation coefficients higher than *R* = 0.996 each time. Despite the complexity of the results obtained for several types of water nozzle internal diameters, the observed waveforms are very similar. The directional coefficients of the lines so defined are about *a* = 2, with a slightly lower one for granite than for limestone. It can be assumed that in the range of pump power up to 30 kW to obtain a cut to a depth of *h* = 1 mm, it takes about 2 kW of pump power.

The feed speed is a quantity that determines the penetration time of the high-pressure abrasive jet in a unit volume of the material being cut. Thus, an increase in this size induces a decrease in the achievable depth of the cut. An illustration of the above can be seen in the charts provided in Figure 11, Figure 12 and Figure 13, for example. These runs have a hyperbolic shape. The power exponents of the functions describing the obtained waveforms are close to minus one, which clearly characterizes the inversely proportional waveforms.

Analyzing these waveforms, it can be concluded that, regardless of the pressures used, the highest results are achieved for a feed speed of *v_p_* = 1 mm·s^−1^. As the jet pressure increases, the gap depths for this parameter are, respectively, *h* = 17.8; 23.5; 44.5; 54.5 and 63.8 mm—that is, the spread of extreme values is as high as 46 mm. For the feed speed *v_p_* = 12 mm·s^−1^ with the change in pressure, depths of h = 1.9; 2.9; 3.4; and 5 were reached, respectively, resulting in a more than 10-fold scatter of just 4.1 mm.

The energy intensity of the process is determined primarily by the power consumed by the high-pressure pump. It can vary in the apparatus used, depending on the pressure values used and the diameter of the water nozzle, from p = 0.6 to 30 kW. The other equipment, i.e., the pre-pump, abrasive feeder and worktable, needs an approximately constant amount of power, amounting to a maximum of p = 1.5 kW. Thus, an assessment aimed at minimizing energy consumption will include only the hydraulic parameters of the structure, treating its other consumption as a constant factor, independent of current processing conditions.

The specific energy of the intersection *E_a_* [59] expressing the energy input required to make an intersection with a unit lateral area of the eroded slot, previously described by Equation (6) was used as the main evaluation index.

The choice of this indicator was determined by its adequacy in a situation of varying both hydraulic parameters and machining conditions. This allows the results presented to be unambiguous, which should give maximum clarity to the analysis.

The energy level of the jet is actually determined by two quantities only. These are the pressure of the liquid and the diameter of the water nozzle. By selecting their respective sizes, the speed of the waterjet, its output and power are determined.

Experimental tests were carried out in accordance with the five-level rotatable experiment plan. Threefold repeatability of the tests was used. This task required the following steps to be conducted [17,60]:Determination of variability range of the studied parameters.Choice of the class of the mathematical model.Coding the analyzed parameters.Gathering the experiment results.Elimination of results with gross mistakes.Calculating the inter-row variance and standard deviation.Checking the homogeneity of variance.Calculating the coefficients of regression function.Statistical analysis of the regression function.Examination of the significance level of the correlation coefficient.Checking the adequacy of the mathematical model.Decoding the regression function.

The average values of the outputs of the object ${\bar{E}}_{a}$ were approximated with a polynomial of the second degree, obtaining the regression equation as two-parameter functions:(12)$${\hat{\bar{E}}}_{a}=b_{o}+b_{1}\cdot \bar{p}+b_{2}\cdot {\bar{d}}_{w}+b_{11}\cdot {\bar{p}}^{2}+b_{22}\cdot {{\bar{d}}_{w}}^{2},\;\left[\mathrm{kJ}\cdot {\mathrm{mm}}^{-2}\right],$$
where: 

$b_{o},\;b_{1},b_{2},\;b_{11}\;\mathrm{and}\;b_{22}$ = unknown coefficients of the regression equation,

${\bar{x}}_{i}$ = input variables: ${\bar{x}}_{1}={\bar{d}}_{w}$ [mm] and ${\bar{x}}_{2}=\bar{p}$ [MPa] or:
${\bar{x}}_{1}={\bar{N}}_{j}$ [kW] and: ${\bar{x}}_{2}={\bar{v}}_{p}$ [mm·s^−1^].

Using matrix calculus, the column vector {b} of the unknown coefficients in Equation (12) was calculated from the matrix formula:(13)$$\left\{b\right\}={\left({\left[\bar{X}\right]}^{T}\left[\bar{X}\right]\right)}^{-1}{\left[\bar{X}\right]}^{T}\left\{\bar{Y}\right\}$$
where:

$\left[\bar{X}\right]$**=** input variable matrix of dimension N × L. For data N = 5 and L = 5, the following matrix forms $\left[\bar{X}\right]$ were developed,

${\left[\bar{X}\right]}^{T}$**=** transposed matrix $\left[\bar{X}\right],$${\left({\left[\bar{X}\right]}^{T}\left[\bar{X}\right]\right)}^{-1}$ = covariance matrix,

$\left\{\bar{Y}\right\}$**=** column vector of the average values of the experimental results.

The boundaries of the confidence region for Regression Function (12) were determined from the following formula:(14)$${\hat{\bar{E}}}_{a}\pm t_{cr\left(\alpha ;f=N-L\right)}\cdot \frac{S_{R}}{\sqrt{N-L-1}}\cdot \sqrt{{\left\{\bar{x}\right\}}^{T}{\left({\left[\bar{X}\right]}^{T}\left[\bar{X}\right]\right)}^{-1}\left\{\bar{x}\right\}},\;\left[\mathrm{kJ}\cdot {\mathrm{mm}}^{-2}\right],$$
where:

${\hat{\bar{E}}}_{a}$= regression equation according to Formula (12),

$t_{cr\left(\alpha ;f\right)}$ = critical value of Student t test for significance level *α* = 0.05 and the number of the degrees of freedom *f* = *N* − *L*,

*N* = number of measurement points in the experimental design: *N* = 13,

*L* = number of unknown coefficients in Regression Equation (12); here, *L* = 5,

$\left\{\bar{x}\right\}$ and ${\left\{\bar{x}\right\}}^{T}$-column vector of the functions of input variables (test factors in real form) and its transposition: ${\left\{\bar{x}\right\}}^{T}=[1\;{\bar{x}}_{1}\;{\bar{x}}_{2}\;{\left({\bar{x}}_{1}\right)}^{2}\;\left({\bar{x}}_{2}{)}^{2}\right]$,

$S_{R}=\sum_{i=1}^{i=N}{\left({\hat{\bar{y}}}_{i}-{\bar{y}}_{i}\right)}^{2}$-residual variance,

${\hat{\bar{y}}}_{i}$ = average values of model outputs for plan points calculated from Equation (12): *(*${\bar{y}}_{i}={\hat{\bar{E}}}_{a}$),

${\bar{y}}_{i}\;$= average values of experimental results.

The test results after statistical processing according to the algorithm presented in the study [17,60] were used to develop regression Equations (15) and (16).

These parameters, in turn, determine the course and efficiency of the mixing of water and abrasive that is, in the end, the value of the energy transferred to the individual abrasive particles determining their micro-cutting abilities. Therefore, Figure 14 just shows the effect of the pressure and internal diameter of the waterjet on the level of the energy consumption of cutting granite with a high-pressure abrasive waterjet. The results obtained are illustrated by a surface with the following equation:(15)$$E_{a}=-1.08^{-7}p^{2}+2.834d_{w}^{2}+7.438^{-5}p-1.436d_{w}+0.248\;\;\left[\mathrm{kJ}\cdot {\mathrm{mm}}^{-2}\right]$$
where: 150 ≤ *p* ≤ 350 MPa; 0.1 ≤ *d_w_*≤ 0.3 mm.

In the considered range of the parameters, it has a clear minimum, running along the value of the inner diameter of the *d_w_* water nozzle of 0.2537 mm. This optimum is quite clear. Decreasing or increasing the inner diameter of the water nozzle results in a noticeable increase in *E_a_*. Intensifying processing by increasing pressure also increases the energy intensity for each nozzle used. At the same time, the observed increase in the energy intensity is mild. For the maximum difference in applied pressures, the increase is only about 0.02 kJ·mm^−2^.

The last issue related to optimization according to the criterion of minimizing the energy intensity is the answer to the question of whether it is more beneficial to intensify the process with an increase in the power of the waterjet or an increase in the erosion time. A graphic illustration of this response can be seen in the example surface shown in Figure 15.

The described tests were carried out for an optimal water nozzle internal diameter of *d_w_* = 0.25 mm. The resulting surface (with the equation shown below):(16)$$E_{a}=-0.001v_{p}^{2}+0.006N_{j}-0.015v_{p}+0.146,\;\left[\mathrm{kJ}\cdot {\mathrm{mm}}^{-2}\right]$$
where: 1 ≤ *v_p_* ≤ 8; mm·s^−1^ and 1 ≤ *N_j_* ≤ 9, kW, reaches the lowest values for the lowest power and feed speed in the limit of *v_p_*=7.5 mm·s^−1^. Intensification by both the increasing power and the decreasing feed speed increases the relative power consumption by a similar amount.

## 4. Conclusions

The research conducted allows us to conclude that when using a high-pressure jet of pure water, sandstone and granite are cut through. A high-pressure abrasive waterjet has a distinctly higher cutting efficiency. Using it, the highest cutting depths were obtained for sandstone. Slightly more difficult to machine is granite and limestone and marble. Hard and fine-grained basalt is the most difficult to cut. A proportional relationship between pressure and depth of the cut was found for all materials tested.

Regardless of the pressures used, the highest cutting depths are achieved for a feed speed of 1 mm·s^−1^. Of course, with higher jet pressure, a significantly greater depth of the cut was obtained. It should be noted that for higher feed values, the differences between the achieved depth of the cut for each pressure value decrease.

To confirm the above conclusions, the study of the effect of jet pressure on the depth of rock cutting should be expanded to the 600 MPa level achieved in state-of-the-art equipment.

In summary, it can be concluded that the least energy-intensive method of cutting is the use, for the highest possible pressure, of an optimal water nozzle internal diameter of *d_w_* = 0.25 mm for the tested pump of about 30 kW. Unfortunately, in this way the maximum capacity of the pump can be reached very quickly, which limits the possibilities of intensification. This entails achieving deeper cuts by increasing erosion time through decreasing the feed speed or using multiple passes. The first option is more favorable because it brings a slight increase in the energy intensity of the cutting process at 35% to a value of *E_a_* = 0.17 kJ·mm^−2^. The use of multiple passages has worse results because the increase in energy intensity is rapid and reaches maximum values of *E_a_* = 0.35 kJ·mm^−2^.

In further work, special attention should be paid to the use of high-pressure waterjets with operating pressures above 420 MPa and even above 600 MPa for cutting rock, as this aspect is still little known.

## Figures and Tables

**Figure 1 materials-16-00647-f001:**
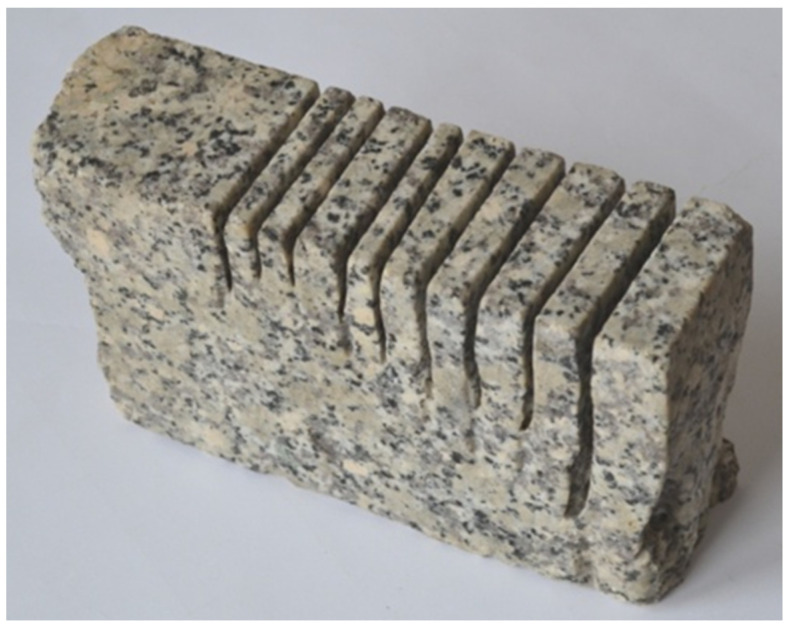
Granite from the Strzegom quarry.

**Figure 2 materials-16-00647-f002:**
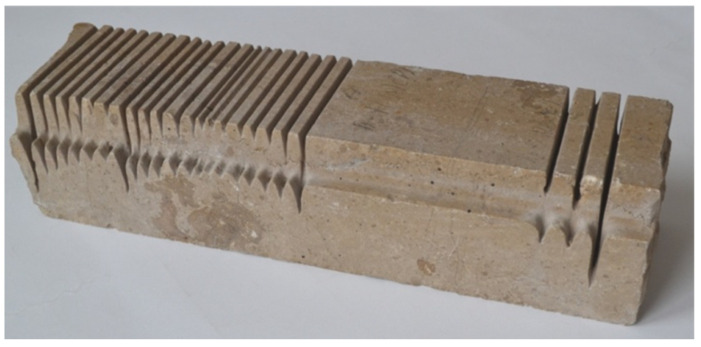
Limestone from Morawica quarry.

**Figure 3 materials-16-00647-f003:**
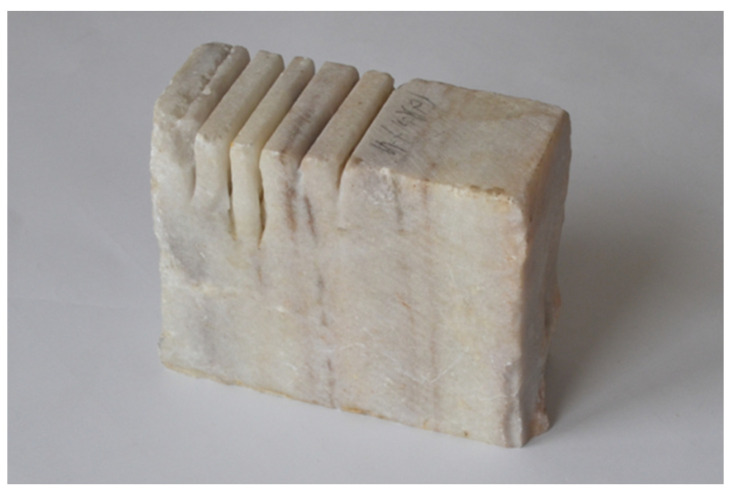
Marble from White Marianna quarry.

**Figure 4 materials-16-00647-f004:**
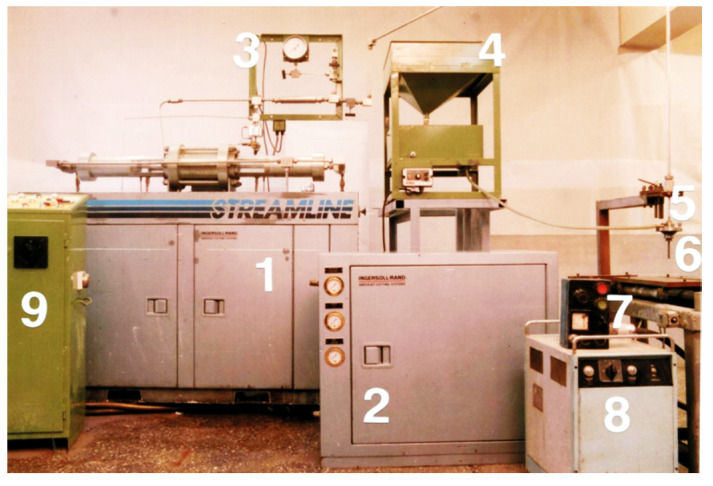
General view of the high-pressure hydroabrasive cutting station: 1—high-pressure pump, 2—pre-pump, 3—measuring panel, 4—abrasive feeder, 5—working head, 6—hydroblasting nozzle, 7—working table, 8—table speed controller, 9—control panel.

**Figure 5 materials-16-00647-f005:**
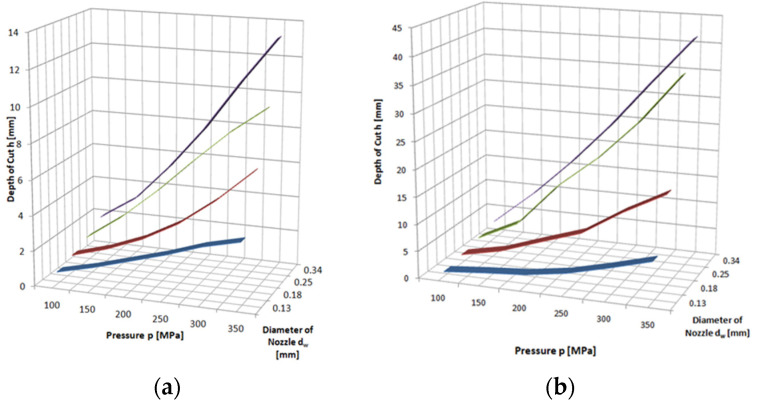
Influence of pressure and waterjet diameter on the depth of the cut of: (**a**) granite and (**b**) sandstone with a pure waterjet: feed speed *v_p_* = 3 mm·s^−1^.

**Figure 6 materials-16-00647-f006:**
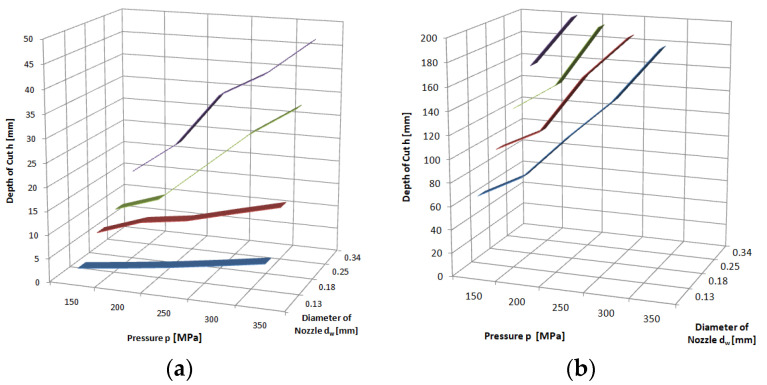
Influence of pressure and waterjet diameter on cutting depth of: (**a**) granite; (**b**) sandstone, with abrasive waterjet: feed speed *v_p_* = 3 mm·s^−1^, abrasive flow rate ${\dot{m}}_{a}=6.7$ g·s^−1^.

**Figure 7 materials-16-00647-f007:**
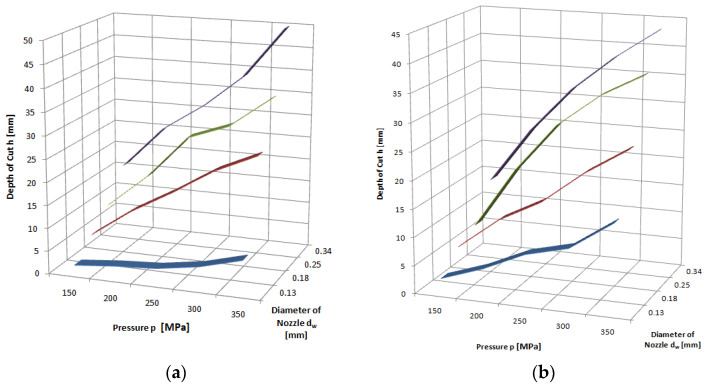
Influence of pressure and waterjet diameter on cutting depth of: (**a**) marble; (**b**) basalt; (**c**) limestone, with abrasive waterjet: feed speed *v_p_* = 3 mm·s^−1^, abrasive flow rate ${\dot{m}}_{a}=6.7$ g·s^−1^.

**Figure 8 materials-16-00647-f008:**
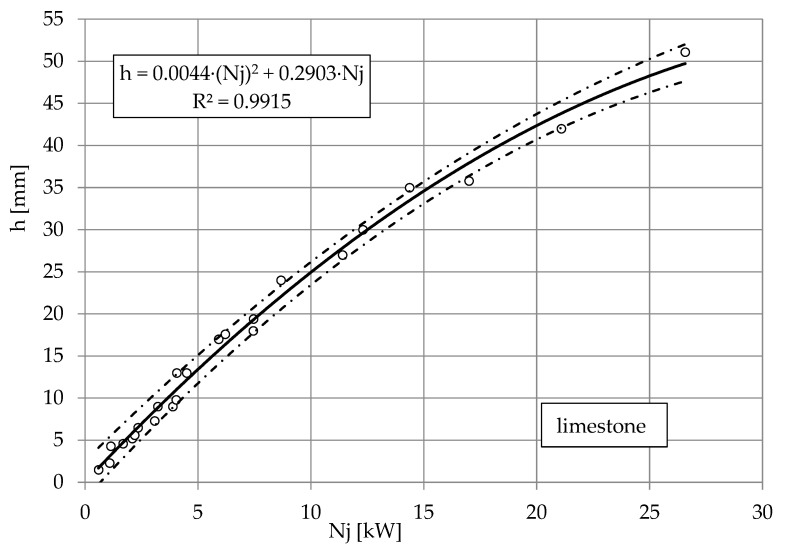
Influence of jet power *Nj* on cutting depth *h* of limestone, with abrasive waterjet: feed speed *v_p_* = 3 mm·s^−1^, abrasive flow rate ${\dot{m}}_{a}=6.7$ g·s^−1^.

**Figure 9 materials-16-00647-f009:**
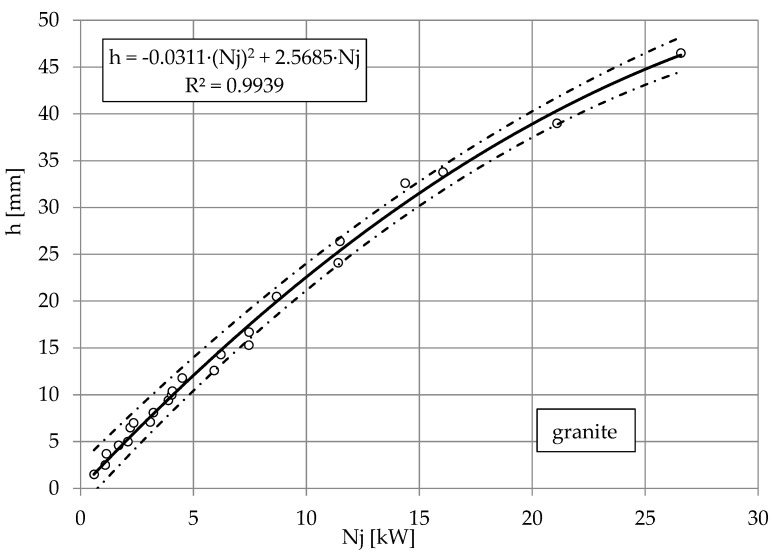
Influence of jet power *Nj* on cutting depth *h* of granite, with abrasive waterjet: feed speed *v_p_* = 3 mm·s^−1^, abrasive flow rate ${\dot{m}}_{a}=6.7$ g·s^−1^.

**Figure 10 materials-16-00647-f010:**
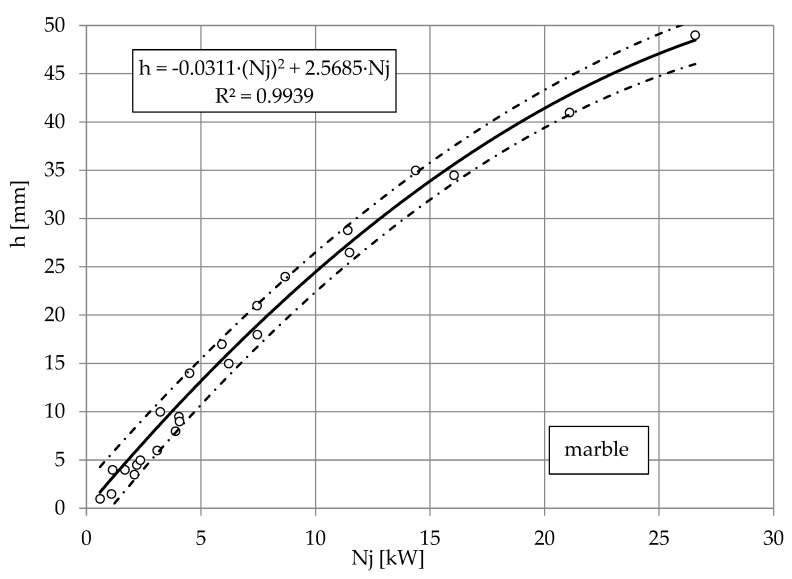
Influence of jet power *Nj* on cutting depth h of marble, with abrasive waterjet: feed speed *v_p_* = 3 mm·s^−1^, abrasive flow rate ${\dot{m}}_{a}=6.7$ g·s^−1^.

**Figure 11 materials-16-00647-f011:**
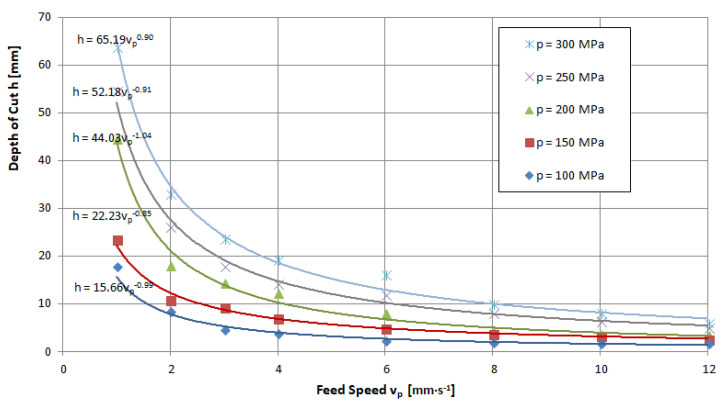
Influence of traverse speed and pressure on the depth of the cut of limestone: nozzle internal diameter *d_w_*=0.25 mm, flow of abrasive ${\dot{m}}_{a}=6.7$ g·s^−1^.

**Figure 12 materials-16-00647-f012:**
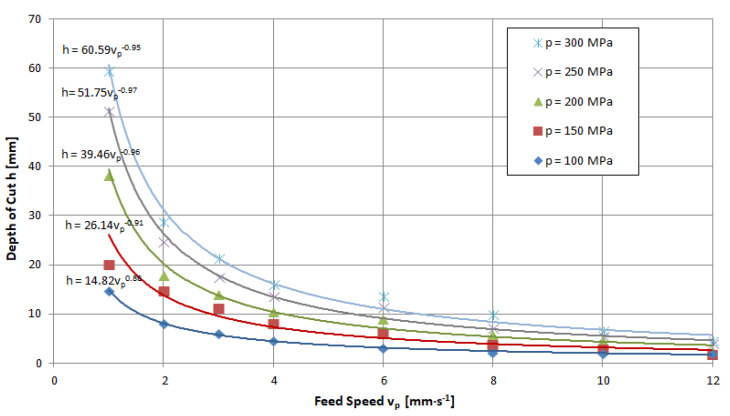
Influence of traverse speed and pressure on the depth of the cut of granite: nozzle internal diameter *d_w_*=0.25 mm, flow of abrasive ${\dot{m}}_{a}=6.7$ g·s^−1^.

**Figure 13 materials-16-00647-f013:**
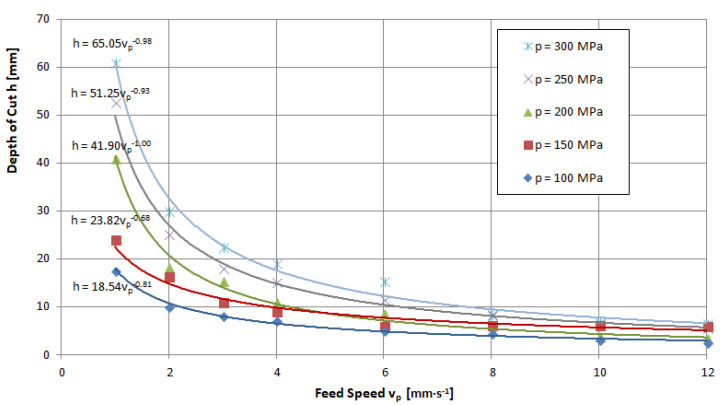
Influence of traverse speed and pressure on the depth of the cut of marble: nozzle internal diameter *d_w_* = 0.25 mm, flow of abrasive ${\dot{m}}_{a}=6.7$ g·s^−1^.

**Figure 14 materials-16-00647-f014:**
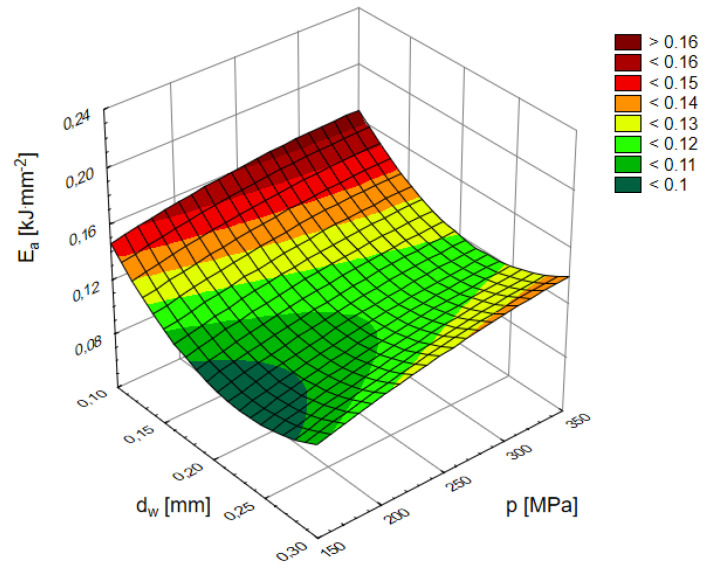
Influence of pressure and waterjet diameter on the energy consumption of granite with abrasive waterjet: feed speed *v_p_* = 3 mm·s^−1^, flow of abrasive ${\dot{m}}_{a}=6.7$ g·s^−1^.

**Figure 15 materials-16-00647-f015:**
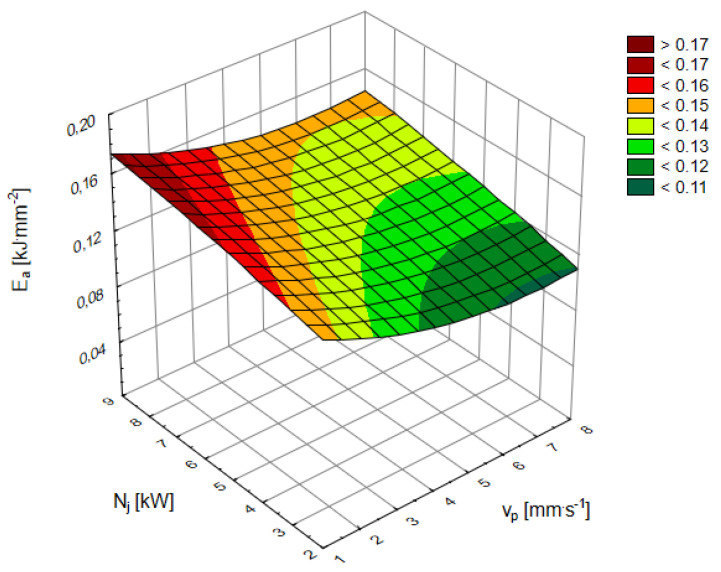
Influence of jet power and traverse speed on the energy consumption of granite with abrasive waterjet: feed speed *v_p_* = 3 mm·s^−1^, flow of abrasive ${\dot{m}}_{a}=6.7$ g·s^−1^.

**Table 1 materials-16-00647-t001:** Division of rocks by geological origin [58].

Group	Magmatic		Sedimentary			Ethamorphic
Type	Deep Rocks	Efflorescent	Clastic Sediments	Organic Sediments	Chemical Sediments	(transformed)
Kind of Rock	granite syenite diorite peridotite	porphyry andesite basalt tuff	breccia sandstone clay slate	limestone dolomite chalk marl	travertine gypsum anhydrite alabaster bauxite	gneiss serpentinite quartzite marble

## Data Availability

Data sharing is not applicable to this article.

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
