# Peer review of "Possibilities of Rock Processing with a High-Pressure Abrasive Waterjet with an Aspect Terms to Minimizing Energy Consumption"

_materials, 2023, doi:10.3390/ma16020647_

Round 1

Reviewer 1 Report

The authors present research into cutting of rock materials by AWJ. They focus on the energy aspects of the cutting process, specifically the minimization of cutting energy. Although the paper presents research under slightly lower parameters (to 380 MPa) than those commonly used today, the paper is interesting and will be able to make the potential reader curious. In future research, I would suggest concentrating on pressures above 420 MPa, and even above 600 MPa, as this aspect is still poorly studied.

In my opinion, the paper is an interesting one and I have no fundamental objections to its content, but I found some errors in this manuscript and it should be improved.

Strength

A great advantage of the paper is the broad analysis of the state of the issue and the large number of sources cited.

Weak

A certain disadvantage of this paper is the mixing of UK English and US English. In research papers, it doesn't matter which version of English you use, but you should use one type consistently. Since US English predominates in the paper, this type should be left. However, a short consultation with an English native speaker would certainly clear up any doubts on this matter.

Noticed errors

1.      Throughout the paper, harmonize the symbol used to separate the integer part from the fractional part of a number written in decimal form. English uses a dot (full stop) rather than a comma (for example numbers in lines: 260, 301, 440)

2.      Throughout the paper, the variables used in the equations should be italicized.

3.      The phrase nozzle diameter is a bit of a mental shortcut. The proper name is nozzle internal diameter, sometimes designated as an acronym: nozzle ID. This would need to be corrected throughout the paper.

4.      Lines 2-4.

Is: Possibilities of rock processing with a high-pressure abrasive water jet with a aspect terms to minimising energy consumption

Should be: Possibilities of rock processing with a high-pressure abrasive water jet with an aspect terms to minimizing energy consumption

5.      Please harmonize your vocabulary: hydro jet (lines 291, 400) or water jet (rest of paper)

6.      Most of the information contained in the Abstract chapter would fit more into the introduction chapter. There is a lack of condensed information about the contents of the article, and this should be improved.

7.      The acronym AWJ, for Abrasive Water Jet, is quite common in the literature. Therefore, I propose to add AWJ to the keywords.

8.      Chapter 2. Materials and Methods does not contain subchapters. To increase clarity, it would be appropriate to reorganized, add several subsections, and move text, such as:

2.1 Cutting efficiency

2.2 Materials

2.3 Methods (or/and test rig). Text from lines 235-245 with Fig.4 would fit perfectly in this subchapter 2.3

9.       Line 135. The unit in equation 5 is incorrect.

10.   Line 366. What is x12?

11.   Chapter 3. Results and Discussion.

To the technical data of the high-pressure pump, in addition to pressure and power, it is worth adding the maximum value of the water flow rate.

12.   Lines 404-412. This paragraph is not truly clear and needs improvement.

13.   Conclusions chapter.

The nine lines of conclusions in the 17-page paper show great restraint of authors in drawing conclusions. In my opinion the potential of the paper is much higher, and the conclusions can be easily extended the authors, with addition of conclusion(s) for further research inclusive.

14.   Literature sources [16] and [23] look the same. Maybe change one to different, for example DOI: 10.3390/ma15113978

Small errors

These errors do not diminish the value of this interesting work, but need to be improved

1.      Line 6. Is: 78-400Szczecinek; should be: 78-400 Szczecinek

2.     Line 81. Is: is the equation M. Hashisha; should be: is the M. Hashish equation

3.      Line 110. Is: by Rehbindera; should be: by Rehbinder

4.      Lines 139, 408. Is: Ea; should be: Ea

5.      Line 214. Is: CaCO3; should be: CaCO3

6.   Line 348-349. One-time use of the experiment plan is completely sufficient.

Author Response

Response Letter to Reviewer 1 Comments

Dear Editors and Reviewers:

Thank you very much for your letter and for the reviewers’ comments concerning our manuscript entitled “Possibilities of rock processing with a high-pressure abrasive water jet with an aspect terms to minimizing energy consumption”.

We have studied the comments carefully and have made corrections which we hope meet with the reviewers’ approvals. Revised portions are marked in green in the paper. The corrections in the paper and the responds to the reviewer’s comments are described below.

Reviewer 1:

Comments and Suggestions for Authors:

The authors present research into cutting of rock materials by AWJ. They focus on the energy aspects of the cutting process, specifically the minimization of cutting energy. Although the paper presents research under slightly lower parameters (to 380 MPa) than those commonly used today, the paper is interesting and will be able to make the potential reader curious. In future research, I would suggest concentrating on pressures above 420 MPa, and even above 600 MPa, as this aspect is still poorly studied.

In my opinion, the paper is an interesting one and I have no fundamental objections to its content, but I found some errors in this manuscript and it should be improved.

Strength

A great advantage of the paper is the broad analysis of the state of the issue and the large number of sources cited.

Weak

A certain disadvantage of this paper is the mixing of UK English and US English. In research papers, it doesn't matter which version of English you use, but you should use one type consistently. Since US English predominates in the paper, this type should be left. However, a short consultation with an English native speaker would certainly clear up any doubts on this matter.

Noticed errors

  1. Throughout the paper, harmonize the symbol used to separate the integer part from the fractional part of a number written in decimal form. English uses a dot (full stop) rather than a comma (for example numbers in lines: 260, 301, 440).
  2. Throughout the paper, the variables used in the equations should be italicized.
  3. The phrase nozzle diameter is a bit of a mental shortcut. The proper name is nozzle internal diameter, sometimes designated as an acronym: nozzle ID. This would need to be corrected throughout the paper.
  4. Lines 2-4.

Is: Possibilities of rock processing with a high-pressure abrasive water jet with a aspect terms to minimising energy consumption

Should be: Possibilities of rock processing with a high-pressure abrasive water jet with an aspect terms to minimizing energy consumption

  1. Please harmonize your vocabulary: hydro jet (lines 291, 400) or water jet (rest of paper)
  2. Most of the information contained in the Abstract chapter would fit more into the introduction chapter. There is a lack of condensed information about the contents of the article, and this should be improved.
  3. The acronym AWJ, for Abrasive Water Jet, is quite common in the literature. Therefore, I propose to add AWJ to the keywords.
  4. Chapter 2. Materials and Methods does not contain subchapters. To increase clarity, it would be appropriate to reorganized, add several subsections, and move text, such as:

2.1 Cutting efficiency

2.2 Materials

2.3 Methods (or/and test rig). Text from lines 235-245 with Fig.4 would fit perfectly in this subchapter 2.3

  1. Line 135. The unit in equation 5 is incorrect.
  2. Line 366. What is x12?
  3. Chapter 3. Results and Discussion.
  4. Lines 404-412. This paragraph is not truly clear and needs improvement.
  5. Conclusions chapter.

The nine lines of conclusions in the 17-page paper show great restraint of authors in drawing conclusions. In my opinion the potential of the paper is much higher, and the conclusions can be easily extended the authors, with addition of conclusion(s) for further research inclusive.

  1. Literature sources [16] and [23] look the same. Maybe change one to different, for example DOI: 10.3390/ma15113978
  2. Small errors

These errors do not diminish the value of this interesting work, but need to be improved

  1. Line 6. Is: 78-400Szczecinek; should be: 78-400 Szczecinek
  2. Line 81. Is: is the equation M. Hashisha; should be: is the M. Hashish equation
  3. Line 110. Is: by Rehbindera; should be: by Rehbinder
  4. Lines 139, 408. Is: Ea; should be: Ea
  5. Line 214. Is: CaCO3; should be: CaCO3
  6. Line 348-349. One-time use of the experiment plan is completely sufficient.

RESPONSE:

Thank you very much for your valuable comments and suggestions.

  1. As suggested by the Reviewer, the article was carefully corrected. Revised portions are market in the green in the text. Throughout the work, commas have been reduced to dots.
  2. All variables used in the equations throughout the paper have been changed and written in italics.
  3. Thank you for your offer. This has been changed throughout the work.
  4. Thank you for your offer. The title of the article has been corrected.
  5. Corrected as suggested by the Reviewer, instead of hydro jet for water jet.
  6. The Abstract chapter has been improved and the information in the Summary chapter has been moved to the introduction chapter.
  7. Keywords has been corrected as suggested by the Reviewer.
  8. In Chapter 2, subchapters have been introduced.
  9. In equation 5, the units have been corrected.
  10. The symbol X12 from the line 366 has been removed. The symbol X12 means the second power. For readability, this symbol in the equations has been corrected to (x1)2 and (x2)2. The meaning of the symbols x1 and x2 is explained above: – input variables:  [mm] and  [MPa] or:  [kW]  and :   [mm·s-1].
  11. Water flow of the high-pressure pump was added.
  12. The information in the lines 404-412 has been corrected.
  13. Thank you for your comments. We've improved the conclusions.
  14. Bibliography was changed.
  15. All small errors have been removed.

In addition, we found some errors in the manuscript and corrected them. The revised contexts are highlighted in the revised manuscript.

Sincerely yours,

Grzegorz Chomka, Maciej Kasperowicz, JarosÅ‚aw Chodór, Jerzy Chudy and Leon KukieÅ‚ka

Reviewer 2 Report

In this paper, the author has studied the application of a high-pressure abrasive water jet in rock processing. The influence of key process parameters (such as pressure, water nozzle diameter, and feed rate) on cutting efficiency is analyzed. The work of the article is obvious and has certain academic research value. However, there are several technical issues that limit the appropriateness of this article for publishing in Materials.

Comprehensive consideration, I would recommend a “Major Revision” of this paper. The following technical issues are to be resolved and may help to improve the quality of the articles:

(1)     The abstract lacks conclusive language. The emphasis on the findings and conclusions of the article is of great help to improve the quality of the article.

(2)     The introduction does have a good description of the literature; the authors provided a lot of information about the current state of the art but should work on focusing the scope of the article in one sentence. A rewrite is necessary to systemically categorize and discuss them. This part must be modified.

(3)     Figures 8 and 9 illustrate the dependence of limestone and granite cutting depth on jet power. Why didn't the author make corresponding research on marble? Please make a supplement or explanation.

(4)     Similar to question 3, the subsequent study did not make any relevant description of marble.

(5)     Figures 12 and 13 lack legends, and be careful to confirm that similar problems exist anywhere else. Please make sure all the formulas are accurate. Tense problems exist in the text.

(6)     From the conclusion, the novelty and main findings of the article should be evident. Despite a great deal of work, however, the conclusions are lacking and need to be supplemented.

Author Response

Response Letter to Reviewer 2 Comments

Dear Editors and Reviewers:

Thank you very much for your letter and for the reviewers’ comments concerning our manuscript entitled “Possibilities of rock processing with a high-pressure abrasive water jet with an aspect terms to minimizing energy consumption”.

We have studied the comments carefully and have made corrections which we hope meet with the reviewers’ approvals. Revised portions are marked in green in the paper. The corrections in the paper and the responds to the reviewer’s comments are described below.

Reviewer 2:

Comments and Suggestions for Authors:

In this paper, the author has studied the application of a high-pressure abrasive water jet in rock processing. The influence of key process parameters (such as pressure, water nozzle diameter, and feed rate) on cutting efficiency is analyzed. The work of the article is obvious and has certain academic research value. However, there are several technical issues that limit the appropriateness of this article for publishing in Materials.

Comprehensive consideration, I would recommend a “Major Revision” of this paper. The following technical issues are to be resolved and may help to improve the quality of the articles:

(1) The abstract lacks conclusive language. The emphasis on the findings and conclusions of the article is of great help to improve the quality of the article.

(2) The introduction does have a good description of the literature; the authors provided a lot of information about the current state of the art but should work on focusing the scope of the article in one sentence. A rewrite is necessary to systemically categorize and discuss them. This part must be modified.

(3) Figures 8 and 9 illustrate the dependence of limestone and granite cutting depth on jet power. Why didn't the author make corresponding research on marble? Please make a supplement or explanation.

(4) Similar to question 3, the subsequent study did not make any relevant description of marble.

(5) Figures 12 and 13 lack legends, and be careful to confirm that similar problems exist anywhere else. Please make sure all the formulas are accurate. Tense problems exist in the text.

(6) From the conclusion, the novelty and main findings of the article should be evident. Despite a great

RESPONSE:

Thank you very much for your valuable comments and suggestions.

(1) The Abstract chapter has been improved.

(2) The Introduction chapter has been modified.

(3) According to the reviewer's proposal, the article was supplemented with marble research (Figure 10).

(4) According to the reviewer's proposal, the article was supplemented with marble research (Figure 13).

(5) The informations in the Figures 12 and 13 (according to the current numbering Figures 14 and 15) have been checked and the drawings have been corrected.

(6) The Conclusion chapter has been modified.

In addition, we found some errors in the manuscript and corrected them. The revised contexts are highlighted in the revised manuscript.

Sincerely yours,

Grzegorz Chomka, Maciej Kasperowicz, JarosÅ‚aw Chodór, Jerzy Chudy and Leon KukieÅ‚ka

Round 2

Reviewer 2 Report

The current version is acceptable.